# Psychometric Properties of the COVID-19 Stress Scale in College Students

**DOI:** 10.3390/bs15101414

**Published:** 2025-10-17

**Authors:** Lynn M. Bielski, Anjolii Diaz, Jocelyn Bolin, Lauren A. Shaffer

**Affiliations:** 1Department of Speech Pathology and Audiology, Ball State University, Muncie, IN 47306, USA; lashaffer@bsu.edu; 2Department of Psychological Sciences, Ball State University, Muncie, IN 47306, USA; adiaz6@bsu.edu; 3Department of Educational Psychology, Ball State University, Muncie, IN 47306, USA; jebolin@bsu.edu

**Keywords:** COVID-19, stress, reliability, college students, mental health

## Abstract

Many experienced isolation and restricted behaviors due to the rapid onset of the COVID-19 pandemic in 2020. Investigations related to the psychological factors such as stress along with the danger of spread and contamination are scarce. The COVID-19 stress scale (CSS) was developed in order to evaluate such stress and anxiety related to the COVID-19 pandemic. The current study investigated the psychometric properties of the CSS, using a survey to provide evidence towards its continued use as a scientifically sound measuring instrument for future acute health crises in a sample of 615 college students (78.80% female, 18.60% male, 1.30% trans male, 1.00% non-binary), with a mean age of 19.10 years. The study partially supported the original measure’s factor structure. The main modification suggests a five-factor structure and removal of items related to less frequently used methods of banking and postal mail. The authors provide suggestions for future validation directions, use of the CSS and development of stress response strategies for students.

## 1. Introduction

In 2020, the United States, among many other countries across the world, experienced an unprecedented time of isolation and restricted behaviors due to a rapidly spreading COVID-19 pandemic. After virtual-only instruction during mid-March to July of 2020, many universities, including our mid-sized Midwestern university, returned to in-person or hybrid style classroom learning experiences and campus activities in Fall 2020. Pandemic levels were still high, causing concern among students, faculty, and staff. Debates ensued over the feasibility of keeping students safe and providing them with the education they paid for. Our university adopted several policies including reduced classroom size, social/physical distancing ([53]), physical barriers, and paper mask dispensers. The university distributed two cloth masks to all students and employees. Mask usage was mandatory on all University properties. A close contact notification system was used to inform students if they needed to quarantine due to contact with the virus. Consequences for not following mask usage and other mandatory safety policies resulted in referral to the Office of the Dean of Students.

The [21] ([21]) urged the population to make efforts to manage stress and protect their mental health during such tumultuous and strenuous times ([39]). Accordingly, researchers began to investigate COVID-19 and its prevention measures’ impact on individuals’ mental well-being. Psychological stress as both a catalyst and the result of serious health concerns or illness is not new ([41]; [47]). Eubank and colleagues examined the impact of COVID-19 stress in students early on, using the Perceived Stress Scale (PSS-12; [22]), and found it psychometrically valid and able to capture the high levels of anxiety and depression in Black, Indigenous, and People of Color (BIPOC) respondents compared to Caucasian respondents. Additional investigations into psychological factors associated with beliefs and behaviors related to spreading and containing viruses are needed to further understand and predict health-related behaviors ([57]). The CSS ([58]) was developed in order to evaluate such stress and anxiety related to the COVID-19.

There are various perspectives to assess the role of stress and its risk of disease. Stress can be discussed biologically as the result of the functioning of specific physiological systems in the body that are particularly responsive to psychologically and physically demanding conditions. Repeated and continuous activation of such systems over time (i.e., sympathetic nervous systems) takes a toll on the body and increases the risk of illness ([40]; [64]). An environmental stress perspective, on the other hand, suggests that our assessments of our surroundings and environmental situations, along with subsequent experiences, can lead to substantial demands. The accumulation of demand from life events, particularly those related to health, change, and uncertainty, is related to several physical conditions ([7]; [34]). However, evaluations of events are subjective, and one’s ability to cope with demands can vary depending on the situation or experiences. This more psychological view of stress emphasizes that if a problem or situation is deemed serious and there is a lack of resources, stress ensues ([36]).

The experience of stress can be viewed through the transactional model ([31]), where individual differences mediate the response to stress. How someone views COVID-19 influences the extent to which they experience stress. In this model the primary appraisal is the perceived threat, and the secondary appraisals are the coping mechanisms available to adequately manage that stress ([30]). Overall, the perceived stress caused by COVID-19 could vary across individuals, and the CSS could be a method of measuring these variations.

Regardless of the perspective subscribed to, COVID-19 can be viewed as a stressful life event if not a traumatic event. A recent study reported the presence of traumatic stress reactions in individuals directly as well as indirectly exposed to the virus ([17]). The pandemic spread rapidly, prompting health and government officials to scramble for resources to combat its spread. Uncertainties also loomed regarding COVID-19 severity, length, and treatment. Events that are beyond one’s control and seem endless are associated with greater stress and reduced immunity ([27]). Indeed, they discuss a shift in immune system functioning from adaptive to potentially detrimental with continued stress exposure. Moreover, COVID-19 resulted in restricted measures such as lockdowns, social/physical distancing, and mask mandates. For the first time in a long time, the US population faced an unpredictable and rapidly progressing situation that confined or restricted their movement both within and outside the country, thereby impeding their ability to engage in typical social activities. Confinement and social isolation are linked to greater experiences of negative emotions such as fear, frustration, anger, and anxiety with potentially long-lasting effects on one’s psychological well-being ([18]; [32]).

Fear of contracting COVID-19 or falling ill, of encountering infected objects or individuals, of dying, of suffering if infected (economically and socially), of loneliness and stigma, as well as feelings of helplessness are common psychological reactions noted in other viral outbreaks (i.e., Ebola, H1N1, MERS) that can intensify with restrictions and mandates enacted to help attenuate the rate of a viral spread ([4]; [56]; [61]). Although empirical evidence links epidemics with stress, investigations into psychological factors associated with beliefs and behaviors related to spreading and containing viruses are scarce ([57]). Such investigations are needed because pandemic-related stressors can exacerbate various types of preexisting mental health problems and increase the risk of anxiety and depression even in individuals with no prior history of mental health disorders ([9]; [43]). Increased stress may also lead to maladaptive behaviors to cope with stress and anxiety, such as self-medicating (i.e., drugs) or self-harm ([25]; [50]). The steadily increasing pandemic research suggests that there will be significant long-term mental health needs ([16]). Data related to anxiety from previous epidemics, such as SARS and H1N1 (swine flu), indicated that anxiety was a key driver of behavior. Too much anxiety resulted in behaviors like over-buying household items, and too little anxiety resulted in less concern for public health measures (i.e., handwashing and vaccination uptake) ([10], [11]; [56]; [59]). These data have helped predict health behaviors and can help plan public health messaging and intervention efforts ([56]). Hence, reliable and valid measurement of COVID-19-related stress is important to further understand its relation to psychological and physical well-being.

In order to assess COVID-19-related stress, [58] ([58]) created the CSS, a 36-item measure consisting of sub-scales of danger and contamination fears, socio-economic consequences, xenophobia, compulsive checking and reassurance seeking, and traumatic stress symptoms. These subscales indicated the extent to which respondents experienced COVID-19-related worries as well as how often respondents engaged in compulsive checking behaviors, or how frequently they experienced problems related to traumatic stress in the past week. While other measures seek to capture fear and anxiety around COVID-19 ([3]; [37]), the CSS captures many dimensions. The items were selected based on previous data ([56]) and consultation with experts on health anxiety. Initial validation of the measure was conducted with population-representative adult samples from Canada and the United States ([58]). Overall, scales performed well on various indices of reliability and validity and offered promise as a tool to understand the types of distress associated with COVID-19 and to identify individuals at risk.

The current study sought to provide further evidence of the CSS psychometric properties in a younger population, college students. [58]’s ([58]) sample’s mean age was close to 50 years of age (M = 49.80; SD = 16.20), and about half of the participants were employed (52.30%). Only 4.4% of the sample were students. The American Institute of Stress ([55]) has described stress among college students as an epidemic spreading across the country, with the [5] ([5]) reporting that three out of four college students self-report feeling stressed. Stress experienced by college students is multi-faceted with a variety of contributing factors, but academic-related stressors are thought to play a major role ([12]; [15]). Beginning and attending college represents a unique developmental stage in a young person’s life ([6]; [49]). This period of emerging adulthood may be a time of increased well-being ([42]), while others experience greater anxiety and depression ([54]). Hence, a college student population is critical to investigate because they may have different concerns and anxieties compared to older and more occupationally diverse populations.

Student burnout, a condition resulting from chronic stress and overwhelming demands ([35]), saw an increase across countries as a result of COVID-19 stressors ([1]). Burnout tends to vary as a function of gender, as girls have been shown to experience higher levels compared to boys ([46]). Similarly to the perception of stress, individuals may experience burnout differently. Overall, burnout has been linked to many negative impacts on health, academic performance, self-esteem, use of positive coping strategies ([33]), and completion of academic program ([14]).

College populations throughout the US may have interpreted stress differently because they were in a setting where mask wearing and other COVID-19 precautions were enforced. The current study examined the factor model of the CSS in a university population with these COVID-19 safety mandates. More research is needed to fully understand the psychological consequences of the pandemic in different populations and, accordingly, formulate psychometrically sound measures to help inform psychological intervention ([62]). The current study investigates the psychometric properties of the CSS to provide evidence towards its continued use as a scientifically sound measuring instrument for future epidemics.

## 2. Materials and Methods

### 2.1. Sample Participants and Data Collection Procedures

An anonymous survey was administered using Qualtrics and distributed across the University to all students, faculty and staff through the Institutional Review Board (IRB) approved recruitment tools (paper flyers and digital campus communication center notifications). Students enrolled in the Psychological Science 100-level course were offered course credit if they completed the survey. No monetary or tangible incentives were offered for those not enrolled in a Psychological Science 100-level course. The survey and project were approved by the Ball State University IRB, protocol number 1668284.

### 2.2. Measures

The CSS is a 36-item measure grouped into five domains, including: (1) danger and contamination (12-items), (2) socio-economic consequences (6-items), (3) xenophobia (6-items), (4) traumatic stress symptoms (6-items), and (5) compulsive checking and reassurance seeking (6-items). Items are rated on a 5-point Likert scale ranging from 0 (not at all) to 4 (extremely) for fear-related items and rated from 0 (never) to 4 (almost always) for frequency-related questions. The scores for each of the five domains are calculated as the sum of ratings for each item in that domain. Table 1 shows the CSS descriptive information and reliability. Higher scores indicate more intense or more frequent perceptions. The English version of the CSS was used in this study with no adaptations or added/removed items.

### 2.3. Analysis Plan

To assess the psychometric properties of the CSS, confirmatory factor analysis (CFA) was performed on the original five-factor structure proposed by [58] ([58]). Data analysis was computed on complete cases (listwise deletion used for missing cases). Model fit statistics, factor loadings, and modification indices were requested. Cronbach’s alpha tests of internal consistency were also performed for each factor. Based on the fit and suggested modifications, subsequent factor structures were tested using exploratory factor analysis. All analyses were run using the R statistical software platform version 4.4.1 ([44]) and the Lavaan package version 0.6-19.

## 3. Results

### 3.1. Sample Characterization

A total of 650 participants took part in the survey, which was available from November 2020 to March 2022. This timeframe was chosen because university precautions for masking, distancing, and changes in class size were in effect. From that total, 35 were removed due to incomplete survey responses, leaving 615 responses in the final data set. The mean age of respondents was 19.10 years, and a majority identified as female (78.80%). The remaining respondents identified as male (18.60%), trans-male (1.30%), and non-binary (1.00%).

### 3.2. Data Analysis

Cronbach’s alpha for the six original subscales of the CSS indicates good internal consistency (danger and contamination = 0.89, contamination = 0.89, socio-economic consequences = 0.92, xenophobia = 0.95, traumatic stress symptoms = 0.89, compulsive checking = 0.83). Results of CFA on the original six-factor structure of the CSS appear in Table 2.

To evaluate model fit, the following criteria for good fit were used: Chi-Square *p* > 0.05. CFI/TLI > 0.90, RMSEA < 0.05.

Model fit statistics indicate poor fit (*Chi-Square* = 1337.70, *p* < 0.01, *CFI* = 0.83, *TLI* = 0.82, *RMSEA* = 0.09). To investigate the poor fit and suggest an alternative structure, modification indices for the original CFA as well as an Exploratory Factor Analysis (EFA) were run on the entire battery of items. The results of the EFA appear in Table 3.

Results of the EFA, in general, supported the structure of the original socio-economic consequences, xenophobia, traumatic stress symptoms, and compulsive checking. Examination of the loadings indicates that the items from the original Contamination factor had generally low to moderate loadings for all items. Items 19, 20, and 21 had only moderate loadings on the original Contamination factor but loaded reasonably well with the items on the original Danger factor (*γ* = 0.67, *γ* = 0.42, *γ* = 0.56). These items assess worry about catching the virus from other people or from things in public spaces.

Modification indices from the original CFA generally agree with the results of the current CFA. Like [58] ([58]), evidence from both the EFA and the modification indices indicated that the contamination factor could be combined with the danger factor to create a five-factor solution. To get the best fit, however, the modification indices also indicated that certain error terms should be correlated that were not noted in the original model (see Table 4). Furthermore, items 22, 23, and 24 did not load onto the danger and contamination factor as previously reported in [58] ([58]). Items 22, 23, and 24, which ask about physically handling currency and mail, loaded well together; however, three items can be considered a bare minimum for the definition of a psychometrically sound factor ([29]). Given the items only loaded well with each other (but to no other factors) and the somewhat antiquated nature of these issues in current culture, the decision was made to delete the items instead of retaining a sixth factor.

### 3.3. Modified Five Factor Structure

The final modified structure contains five factors. Cronbach alphas for the modified five subscales all indicate good internal consistency: danger and contamination (γ = 0.92), socio-economic consequences (*γ* = 0.92), xenophobia (*γ* = 0.95), traumatic stress symptoms (*γ* = 0.89) and compulsive checking (*γ* = 0.83). Based on the results of the EFA and modification indices, items 22, 23 and 24 were removed entirely from the model. In addition to the item changes, modification indices recommended correlating errors between multiple pairs of variables (See Table 5). CFA results appear in Table 3. Fit for the modified five factor structure was good (*Chi-Square* = 801.90, *p* < 0.01, *CFI* = 0.92, *TLI* = 0.91, *RMSEA* = 0.07).

## 4. Discussion

The proposed factor structure of the original 36-item CSS consisted of a five-factor solution that included the following: (1) danger and contamination fears, (2) fears about economic consequences, (3) xenophobia, (4) compulsive checking and reassurance seeking, and (5) traumatic stress symptoms about COVID-19. The average age of the survey respondents was almost 50 years old, and the majority had completed a full or partial college degree. The creators of the CSS ([58]) have called for additional research to expand the breadth of content of the measure. The aim of the current study was to evaluated the CSS’s fit in a college student sample returning to in-person instruction in the Fall of 2020. All students, faculty, and staff were required to use masks while in University or University-associated settings and to maintain physical distancing ([53]) in classes and University-associated gatherings. Consequences were also associated with not abiding by these university COVID-19 policies.

Although not many, published psychometric studies of the CSS report mixed evidence, either suggesting a 5-item or 6-item solution depending on the population (i.e., [2]; [38]). Results partially supported the original five-factor model. The authors recommend considering reducing the 36-item survey to 33 items and slightly modifying one of the scales. Items assessing one’s worry and fears still fit well under pandemic danger and contamination. However, the CFA fit and loading indices indicated the need to reorganize and remove several items from this scale for an acceptable model fit. Items assessing the risk of catching the virus from handling money or mail and engaging in cash transactions may be less prevalent now than in the past; however, this may be a function of age and country. Even before COVID-19, mobile payments and debit cards surpassed cash as the most prevalent payment method used by US college students ([48]). Although the proportion of Americans whose typical week includes mostly cash purchases has steadily decreased from 18% in 2018 to 14% in 2022 ([24]), this trend of cashless payments may not hold up in countries other than the US, as cash usage is variable internationally ([8]). A study examining a cross-section of college students reported that postal mail was one of the least preferred methods of communication, with email and texting being most preferred for academic and non-academic communications ([52]). Individuals 35 years of age and under also send and receive fewer pieces of postal mail and tend to adopt newer and faster communication technologies ([60]); however, postal service usage may vary across age groups and countries ([28]).

This study is one of the first large-scale validations of mainly college students in the United States (US). It also provides support for the psychometric structure of the CSS. Some limitations of this study include reliance on a convenience sample, absence of test–retest reliability and lack of measurement invariance testing. Further studies are also required to support the construct validity of the CSS, particularly as it relates to post COVID-19 life and reliance on digital communication. The CSS, nevertheless, offers promise as a tool for better understanding the distress associated with pandemics and for identifying people in need of mental health services. Given that approximately 80% of college students report feeling stressed sometimes or often, and over 70% of American college students report a decline in mental health during the pandemic ([55]; [51]), psychometrically sound measures are vital for continued research and assessment improvement. Psychological issues can have a major impact on college students, including drop-out, poor health outcomes and lower job placement ([26]).

It is important to note that the study’s sample primarily consisted of freshmen transitioning into a Midwestern college with COVID-19 imposed restrictions on their learning and socialization. A shortened 33 item CSS may allow researchers to better capture the most critical issues related to each factor for different populations, and under different conditions. The data in the present study were gathered while mandates like mask wearing were in place on campus. Health mandates in response to COVID-19 varied across countries ([63]) and across states in the US ([65]), and these factors may have an impact on the dimensions captured by the CSS.

It should also be noted that the sample used in this study was very imbalanced in terms of gender. In addition to being an issue of sample generalizability, this imbalance made checking measurement invariance across genders not possible due to the small number of males included in the study. A sample with greater balance across genders would be beneficial because previous studies have shown that stress appraisal varies, with women reporting higher levels of stress compared to men ([13]). It is possible that gender-based differences in stress appraisal had an impact on the current data set, considering 78.80% of the respondents were female.

Future investigations on the CSS’s psychometrics are needed to better understand its item and factor structure in different contexts, as all college campuses are not homogeneous. For example, a survey of college students attending a university in the Bronx indicated female-identifying BIPOC students showed higher resilience compared to students of other genders/races ([20]). A previous study compared the experiences during the pandemic in college student in two countries, Argentina and the US ([45]) and found differences in COVID-related fear and stress between countries. The CSS could also be used to explore differences across counties and age groups and the implications for social support and coping strategies.

## 5. Conclusions

The current study examined the psychometric properties of the CSS in a US college sample. The CSS continues to show promise as a stress-measuring instrument for future epidemics and would benefit from additional cross-cultural and longitudinal validation. Future evaluations of the CSS and its findings may provide valuable stress measures and be critical in developing and providing student mental health support services and activities that support mental health, reduce stress and address student burnout ([19]; [23]). Although the current study suggests the removal of CSS items dealing with cash transactions and use of postal mail, additional research is needed to examine its breadth of content and validity across other age groups and countries.

## Figures and Tables

**Table 1 behavsci-15-01414-t001:** CSS Descriptive Information.

Item		N	Min	Max	Mean	Std. Dev	* Alpha If Deleted
CSS1_1	I am worried about catching the virus	156	0	4	1.78	1.29	0.88
CSS1_2	I am worried that basic hygiene is not enough to keep me safe from the virus	156	0	4	1.65	1.27	0.89
CSS1_3	I am worried that our healthcare system is unable to keep me safe from the virus	156	0	4	1.71	1.35	0.88
CSS1_4	I am worried that I can’t keep my family safe from the virus	156	0	4	2.22	1.42	0.88
CSS1_5	I am worried that our healthcare system won’t be able to protect my loved ones	155	0	4	2.15	1.43	0.87
CSS1_6	I am worried that social distancing is not enough to keep me safe from the virus	156	0	4	1.78	1.38	0.87
CSS1_7	I am worried about grocery stores running out of food	156	0	4	0.73	0.99	0.90
CSS1_8	I am worried about grocery stores running out of cold or flu remedies	156	0	4	0.86	1.01	0.89
CSS1_9	I am worried about pharmacies running out of prescription medicines	156	0	4	0.87	1.04	0.90
CSS1_10	I am worried about grocery stores running out of water	156	0	4	0.87	1.07	0.91
CSS1_11	I am worried about grocery stores running out of cleaning or disinfectant supplies	156	0	4	1.40	1.25	0.92
CSS1_12	I am worried that grocery stores will close down	155	0	4	0.89	1.11	0.92
CSS1_13	I am worried that foreigners are spreading the virus in my country	156	0	4	0.40	0.82	0.95
CSS1_14	If I met a person from a foreign country, I’d be worried that they might have the virus	156	0	4	0.49	0.86	0.94
CSS1_15	I am worried about coming into contact with foreigners because they might have the virus	156	0	4	0.38	0.81	0.94
CSS1_16	I am worried that foreigners are spreading the virus because they’re not as clean as we are	156	0	4	0.20	0.69	0.94
CSS1_17	If I went to a restaurant that specialized in foreign foods, I’d be worried about catching the virus	156	0	4	0.22	0.72	0.95
CSS1_18	If I was in an elevator with a group of foreigners, I’d be worried that they’re infected with the virus	156	0	4	0.32	0.77	0.94
CSS1_19	I am worried that people around me will infect me with the virus	156	0	4	1.47	1.08	0.91
CSS1_20	I am worried that if I touched something in a public space I would catch the virus	156	0	4	1.33	1.11	0.89
CSS1_21	I am worried that if someone coughed or sneezed near me, I would catch the virus	156	0	4	1.72	1.28	0.89
CSS1_22	I am worried that I might catch the virus from handling money or using a debit machine	156	0	4	1.13	1.17	0.88
CSS1_23	I am worried about taking change in cash transactions	156	0	4	0.97	1.11	0.89
CSS1_24	I am worried that my mail has been contaminated by mail handlers	156	0	4	0.58	0.87	0.89
CSS2_1	I had trouble sleeping because I worried about the virus	156	0	4	0.41	0.79	0.86
CSS2_2	I had bad dreams about the virus	156	0	4	0.28	0.68	0.88
CSS2_3	I thought about the virus when I didn’t mean to	155	0	4	1.05	1.24	0.89
CSS2_4	Disturbing mental images about the virus popped into my mind against my will	155	0	4	0.45	0.88	0.85
CSS2_5	I had trouble concentrating because I kept thinking about the virus	156	0	4	0.56	0.92	0.84
CSS2_6	Reminders of the virus caused me to have physical reactions, such as sweating or a pounding heart	156	0	4	0.35	0.83	0.86
CSS3_1	Social media posts concerning COVID-19	156	0	4	1.53	1.20	0.81
CSS3_2	YouTube videos about COVID-19	155	0	4	0.73	0.99	0.82
CSS3_3	Seeking reassurance from friends or family about COVID-19	156	0	4	1.12	1.12	0.78
CSS3_4	Checking your own body for signs of infection (e.g., taking your temperature)	156	0	4	1.79	1.26	0.81
CSS3_5	Asking health professionals (e.g., doctors or pharmacists) for advice about COVID-19	156	0	4	0.60	0.85	0.80
CSS3_6	Searched the Internet for treatments for COVID-19	156	0	4	0.83	1.13	0.78

* Cronbach alpha if deleted from original subscale (shaded). Original Cronbach alphas: Danger (1_1-1_6) = 0.89; SocEcon (1_7-1_12) = 0.92; Xeno (1_13-1_18) = 0.95; Contamination (1_19-1_24) = 0.91; TS (2_1-2_6) = 0.88; Checking (3_1-3_6) = 0.83.

**Table 2 behavsci-15-01414-t002:** Confirmatory Factor Analysis Solution for Original 6 CSS Subscales.

Danger		SocEcon		Xenophobia		Contamination		TS		Checking	
CSS1_1	0.77 *	CSS1_7	0.86 *	CSS1_13	0.79 *	CSS1_19	0.64 *	CSS2_1	0.79 *	CSS3_1	0.59 *
CSS1_2	0.71 *	CSS1_8	0.90 *	CSS1_14	0.87 *	CSS1_20	0.78 *	CSS2_2	0.57 *	CSS3_2	0.49 *
CSS1_3	0.75 *	CSS1_9	0.87 *	CSS1_15	0.92 *	CSS1_21	0.70 *	CSS2_3	0.69 *	CSS3_3	0.78 *
CSS1_4	0.74 *	CSS1_10	0.81 *	CSS1_16	0.92 *	CSS1_22	0.92 *	CSS2_4	0.86 *	CSS3_4	0.66 *
CSS1_5	0.79 *	CSS1_11	0.77 *	CSS1_17	0.88 *	CSS1_23	0.89 *	CSS2_5	0.92 *	CSS3_5	0.67 *
CSS1_6	0.80 *	CSS1_12	0.75 *	CSS1_18	0.92 *	CSS1_24	0.83 *	CSS2_6	0.74 *	CSS3_6	0.79 *

Note. SocEcon = socio-economic consequences TS = traumatic stress symptoms; *Chi-Square* = 1337.7 *p* < 0.001, Root mean square error of approximation (*RMSEA*) = 0.09, Comparative fit index (*CFI*) = 0.83, Tucker–Lewis index (*TLI*) = 0.82.* factor loading significant *p* < 0.001.

**Table 3 behavsci-15-01414-t003:** EFA Exploratory Factor Loadings for Promax Rotated Solution, with Eigenvalues greater than 1.

	Factor Loading						
	1	2	3	4	5	6	7
Factor 1 **Xenophobia**							
CSS_1_13: I am worried that foreigners are spreading the virus in my country	**0.75**	0.04	0.17	0.19	0.07	0.18	−0.08
CSS_1_14: If I met a person from a foreign country, I’d be worried that they might have the virus	**0.84**	0.01	0.19	0.07	0.08	0.18	−0.02
CSS_1_15: I am worried about coming into contact with foreigners because they might have the virus	**0.89**	0.05	0.14	0.07	0.11	0.12	−0.04
CSS_1_16: I am worried that foreigners are spreading the virus because they’re not as clean as we are	**0.91**	0.03	0.06	0.08	0.01	0.03	0.02
CSS_1_17: If I went to a restaurant that specialized in foreign foods, I’d be worried about catching the virus	**0.89**	0.02	−0.03	0.09	0.05	−0.02	0.06
CSS_1_18: If I was in an elevator with a group of foreigners, I’d be worried that they’re infected with the virus	**0.92**	0.02	0.01	0.12	0.08	−0.01	0.04
Factor 2 **Danger**							
CSS_1_1: I am worried about catching the virus	0.06	**0.70**	0.06	0.08	0.29	0.26	0.15
CSS_1_2 I am worried that basic hygiene (e.g., handwashing) is not enough to keep me safe from the virus	0.04	**0.62**	0.12	−0.04	0.33	0.12	0.02
CSS_1_3: I am worried that our healthcare system is unable to keep me safe from the virus	−0.02	**0.67**	0.16	0.22	0.19	0.03	−0.05
CSS_1_4: I am worried that I can’t keep my family safe from the virus	0.03	**0.68**	0.29	0.17	0.05	0.09	−0.01
CSS_1_5: I am worried that our healthcare system won’t be able to protect my loved ones	0.03	**0.77**	0.25	0.25	−0.03	0.06	−0.08
CSS_1_6: I am worried that social distancing is not enough to keep me safe from the virus	−0.050	**0.75**	0.14	0.09	0.21	0.09	−0.01
Factor 3 **SocEcon**							
CSS_1_7: I am worried about grocery stores running out of food	0.06	0.24	**0.81**	0.07	0.18	0.05	0.01
CSS_1_8: I am worried about grocery stores running out of cold or flu remedies	0.09	0.20	**0.84**	0.09	0.13	0.12	−0.01
CSS_1_9: I am worried about pharmacies running out of prescription medicines	0.10	0.18	**0.81**	0.08	0.19	0.15	−0.15
CSS_1_10: I am worried about grocery stores running out of water	0.01	0.13	**0.79**	0.08	0.06	0.15	0.03
CSS_1_11: I am worried about grocery stores running out of cleaning or disinfectant supplies	−0.02	0.19	**0.72**	0.15	0.12	0.12	0.17
CSS_1_12: I am worried that grocery stores will close down	0.20	0.11	**0.71**	0.09	0.08	0.13	0.04
Factor 4 **TS**							
CSS_2_1: I had trouble sleeping because I worried about the virus	0.08	0.22	0.07	**0.67**	0.13	0.32	−0.05
CSS_2_2: I had bad dreams about the virus	0.27	0.11	0.13	**0.52**	−0.07	0.16	0.09
CSS_2_3: I thought about the virus when I didn’t mean to	0.05	0.26	−0.01	**0.63**	0.08	0.16	0.46
CSS_2_4: Disturbing mental images about the virus popped into my mind against my will	0.10	0.11	0.01	**0.83**	0.19	0.09	0.01
CSS_2_5: I had trouble concentrating because I kept thinking about the virus	0.05	0.19	0.11	**0.87**	0.18	0.12	0.05
CSS_2_6: Reminders of the virus caused me to have physical reactions, such as sweating or a pounding heart	0.25	0.17	0.18	**0.66**	0.09	0.24	−0.09
CSS_3_1: Social media posts concerning COVID-19	−0.12	0.30	0.18	0.23	0.15	**0.47**	0.43
Factor 5 **Contamination**							
CSS_1_19: I am worried that people around me will infect me with the virus	0.10	**0.67**	0.16	0.29	0.29	0.06	0.27
CSS_1_20: I am worried that if I touched something in a public space (e.g., handrail, door handle), I would catch the virus	0.03	**0.42**	0.18	0.08	0.63	0.13	0.15
CSS_1_21: I am worried that if someone coughed or sneezed near me, I would catch the virus	0.15	**0.56**	0.19	0.24	0.39	0.06	0.24
CSS_1_22: I am worried that I might catch the virus from handling money or using a debit machine	0.11	0.29	0.15	0.14	**0.85**	0.10	0.08
CSS_1_23: I am worried about taking change in cash transactions	0.07	0.29	0.20	0.17	**0.80**	0.07	−0.01
CSS_1_24: I am worried that my mail has been contaminated by mail handlers	0.23	0.26	0.28	0.18	**0.71**	0.16	−0.14
Factor 6 **Checking**							
CSS_3_2: YouTube videos about COVID-19	0.02	0.01	0.15	0.11	0.08	**0.55**	0.12
CSS_3_3: Seeking reassurance from friends or family about COVID-19	0.13	0.15	0.19	0.33	0.21	**0.59**	0.19
CSS_3_4: Checking your own body for signs of infection (e.g., taking your temperature)	0.13	0.34	0.29	0.24	0.03	**0.39**	0.13
CSS_3_5: Asking health professionals (e.g., doctors or pharmacists) for advice about COVID-19	0.19	0.12	0.14	0.14	0.06	**0.70**	−0.23
CSS_3_6: Searched the Internet for treatments for COVID-19	0.20	0.21	0.10	0.38	0.04	**0.65**	−0.05

Note. SocEcon = socio-economic consequences TS = traumatic stress symptoms; *RMSEA* = 0.09, *CFI* = 0.83, *TLI* = 0.82. Items loading to each factor (1–7) are bolded.

**Table 4 behavsci-15-01414-t004:** Confirmatory Factor Analysis Solution for Modified 5 Factor CSS Solution.

Danger		SocEcon		Xenophobia		TS		Checking	
CSS1_1	0.83 *	CSS1_7	0.86 *	CSS1_13	0.78 *	CSS2_1	0.79 *	CSS3_1	0.63 *
CSS1_2	0.72 *	CSS1_8	0.90 *	CSS1_14	0.85 *	CSS2_2	0.57 *	CSS3_2	0.49 *
CSS1_3	0.69 *	CSS1_9	0.87 *	CSS1_15	0.89 *	CSS2_3	0.69 *	CSS3_3	0.80 *
CSS1_4	0.67 *	CSS1_10	0.81 *	CSS1_16	0.91 *	CSS2_4	0.86 *	CSS3_4	0.66 *
CSS1_5	0.67 *	CSS1_11	0.77 *	CSS1_17	0.88 *	CSS2_5	0.92 *	CSS3_5	0.59 *
CSS1_6	0.79 *	CSS1_12	0.75 *	CSS1_18	0.94 *	CSS2_6	0.75 *	CSS3_6	0.73 *
CSS1_19	0.81 *								
CSS1_21	0.70 *								
CSS1_20	0.66 *								

Note. SocEcon = socio-economic consequences TS = traumatic stress symptoms; Chi-square = 801.90, *p* < 0.01, *RMSEA* = 0.07, *CFI* = 0.92, *TLI* = 0.91. * factor loading significant *p* < 0.001.

**Table 5 behavsci-15-01414-t005:** Correlated Error Terms in 5 Factor CFA.

Factor	Items
Danger	CSS_1_3, CSS_1_5
	CSS_1_4, CSS_1_5
	CSS_1_19, CSS_1_21
	CSS_1_20, CSS_1_21
Xenophobia	CSS_1_13, CSS_1_14
	CSS_1_14, CSS_1_15
	CSS_1_16, CSS_1_1
TS, Checking	CSS_2_3, CSS_3_1
Checking	CSS_3_5, CSS_3_6

Note. TS = traumatic stress symptoms.

## Data Availability

Data available on request.

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
