# Peer review of "Psychometric Properties of the COVID-19 Stress Scale in College Students"

_behavsci, 2025, doi:10.3390/bs15101414_

Round 1
Reviewer 1 Report (Previous Reviewer 2)
Comments and Suggestions for Authors
Thank you for taking the time to address the recent comments and responding appropriately. I think the revisions add improved context and make this a much stronger manuscript.
Author Response
|
Response to Reviewer 1 Comments
|
|||
|
|
|
|
|
|
|
|||
|
Questions for General Evaluation |
Reviewer 1 Evaluation |
|
|
|
Is the content succinctly described and contextualized with respect to previous and present theoretical background and empirical research (if applicable) on the topic? |
Yes |
|
|
|
Are the research design, questions, hypotheses and methods clearly stated? |
Yes |
|
|
|
Are the arguments and discussion of findings coherent, balanced and compelling? |
Yes |
|
|
|
For empirical research, are the results clearly presented? |
Yes |
|
|
|
Is the article adequately referenced? |
Yes |
|
|
|
Are the conclusions supported by the results presented in the article or referenced in secondary literature?
|
Yes |
|
|
|
Comments and Suggestions for Authors |
|||
|
Thank you for taking the time to address the recent comments and responding appropriately. I think the revisions add improved context and make this a much stronger manuscript.
|
|||
|
Response: Thank you for your time and constructive feedback on our manuscript.
|
|||
|
|
|||
|
|
|||
|
|
|
||
|
|
|
||
|
|
|
|
|
|
|
|
|
|
|
|
|
|
||
|
|
|
|
|
|
||
|
|
||
|
|
||
|
|
||
|
|
||
Reviewer 2 Report (New Reviewer)
Comments and Suggestions for Authors
Dear authors,
Here are some possibilities to improve your paper:
- Please divide the Introduction part in three or for parts and put your aim of the resesarch in the last one. Or let it stay like this but make them more connected. This way, the connection between pasusses is missing.
- You omitted descriptive statistics for each item and subscales (M, SD, Cronbach's alphas an Cronbach's alpha if item is deleted from scale/subscale). For both versions of the scale.
- The conclusion is inadequate an short. It should be completely rewritten.
- The Discussion is not good. Please consider some references, especially about use of term "social distance". It is been completely wrong. Also, you should pay more attention on future directions of the research and weakness of the manuscript. For example, it would be helpfull if you make comparation according to sex, or something else, that can give you chance to better connect yours results with other papers.
Reccomendations
Šuvaković. U. 2020. in journal Sociološki pregled about wrong use of term "social distance" in COVID-19 pandemic situation.
I could reccomend some more references if you wish.
Author Response
Response to Reviewer 2 Comments
|
|
|
|
|
2. Questions for General Evaluation |
Evaluation |
|
|
Is the content succinctly described and contextualized with respect to previous and present theoretical background and empirical research (if applicable) on the topic? |
Must be improved |
|
|
Are the research design, questions, hypotheses and methods clearly stated? |
Must be improved |
|
|
Are the arguments and discussion of findings coherent, balanced and compelling? |
Must be improved |
|
|
For empirical research, are the results clearly presented? |
Must be improved |
|
|
Is the article adequately referenced? |
Can be improved |
|
|
Are the conclusions supported by the results presented in the article or referenced in secondary literature?
|
Can be improved |
|
Comments and Suggestions for Authors
Here are some possibilities to improve your paper:
Please divide the Introduction part in three or for parts and put your aim of the research in the last one. Or let it stay like this but make them more connected. This way, the connection between pauses is missing.
Response: Thank you. The Introduction has been revised and the research aim is stated at the end of the section.
You omitted descriptive statistics for each item and subscales (M, SD, Cronbach's alphas an Cronbach's alpha if item is deleted from scale/subscale). For both versions of the scale.
Response: Thank you. Cronbach alphas are reported for both versions of the scale in text at the beginning of section 3.2 Data Analysis, section 3.4 Modified Five Factor Structure. An additional table (Table 1) has been added with item descriptive information.
The conclusion is inadequate an short. It should be completely rewritten.
Response: Thank you. The conclusions section has been revised.
The Discussion is not good. Please consider some references, especially about use of term "social distance". It is been completely wrong. Also, you should pay more attention on future directions of the research and weakness of the manuscript. For example, it would be helpful if you make comparation according to sex, or something else, that can give you chance to better connect yours results with other papers.
Response: Thank you. The term social distance has been revised in the paper to include physical distancing, and the recommended reference has been added. The discussion section has been revised to include a discussion of stress appraisal differences based on sex, and country of data collection. Additional references added. Information on the weaknesses of the study were also added to the discussion.
Reviewer 3 Report (New Reviewer)
Comments and Suggestions for Authors
Thank you for the opportunity to review the manuscript entitled “Psychometric Properties of the COVID-19 Stress Scale in College Students”. The study aims to validate the COVID-19 Stress Scale (CSS) in a sample of U.S. college students.
The manuscript is well-structured, employs appropriate analyses (CFA/EFA), and provides useful insights into the scale’s psychometric performance. However, several methodological, theoretical, and editorial issues need to be addressed before the paper can be considered for publication.
Abstract
1) The abstract is clear but too general. Please, report demographic details (mean age, gender distribution), briefly state the main modification (5-factor solution with item deletions), and replace vague statements (“authors provide future recommendations”) with more specific implications.
Introduction
2) The introduction situates the study well within the pandemic context but contains redundancies and some awkward phrasing. Streamlining is recommended.
3) The theoretical framing of stress should be strengthened by explicitly linking transactional stress theory (Lazarus & Folkman) and psychological measurement theory to the rationale for scale validation.
4) The rationale for focusing on college students could be better justified using literature on student stress as a distinct phenomenon (e.g., academic stress, developmental stage).
5) The framing could be strengthened by including more recent comparative international data on staff distress or burnout to highlight the gap this study fills. Indeed, burnout is a multi-component construct formed by different and complex size. It is also necessary to refer and comment on the authors' results in relation to the change that the theoretical framework on burnout has undergone in recent years. Furthermore, it is necessary to integrate the literature review, also referring not only to your own national context, but also international literature. Here are some recent works that suit your theme and which I think may be useful for expanding and updating this section:
- Fiorilli, C., Barni, D., Russo, C., Marchetti, V., Angelini, G., & Romano, L. (2022). Students’ Burnout at University: The Role of Gender and Worker Status. International Journal of Environmental Research and Public Health, 19(18), 11341.
- Liu, C., & Ma, J. (2020). Social media addiction and burnout: The mediating roles of envy and social media use anxiety. Current Psychology, 39(6), 1883-1891.
- Salmela-Aro, K., Upadyaya, K., Hakkarainen, K., Lonka, K., & Alho, K. (2017). The dark side of internet use: Two longitudinal studies of excessive internet use, depressive symptoms, school burnout and engagement among Finnish early and late adolescents. Journal of youth and adolescence, 46, 343-357.
Expanding the literature review beyond a national context by incorporating international perspectives would provide a more comprehensive understanding of the topic.
Methods
6) The sample è adequately described, but response rate and recruitment beyond the introductory psychology course should be clarified. Were participants representative of the wider student body?
7) The description of the CSS is adequate but incomplete. Please add:
- Example items for each domain.
- Reliability indices (Cronbach’s alpha) for subscales in this sample.
- Whether the original English version was used or if any adaptation occurred.
Data Analysis
8) CFA/EFA are appropriate, but the manuscript should clearly state:
- Cut-off thresholds for RMSEA, CFI, TLI considered acceptable.
- Whether measurement invariance across gender was tested (important given the gender imbalance in the sample).
- How missing data were handled.
9) Subsections “Participants,” “Measures,” “Data Analysis”) should be clearly separated for readability and to align with reporting standards.
Results
10) Results are presented in detail, but tables should conform to APA style (consistent decimal places, italicized statistical symbols).
11) The initial CFA model shows poor fit (CFI = .83, RMSEA = .09). The rationale for item removal (e.g., cash/mail items) needs to be more transparent — was this a purely statistical decision, or also conceptually justified (low relevance for young adults)?
12) Item-level results (means, SDs, factor loadings, inter-factor correlations) should be summarized more clearly to allow readers to evaluate construct validity.
13) Reporting effect sizes and reliability indices alongside model fit would strengthen the psychometric evidence.
Discussion
14) The discussion acknowledges the modifications to the CSS but should better emphasize the novel contribution: this is one of the first large-scale validations in U.S. college students.
15) Comparisons with international CSS validations are too brief. A more systematic discussion is needed to explain why the contamination/cash/mail items failed in this context.
16) Practical implications should be expanded:
- How can university counseling centers or student support services use the CSS?
- What are the advantages of a shortened 33-item CSS for applied research and practice?
17) Limitations should include: reliance on a convenience sample, absence of test–retest reliability, and lack of measurement invariance testing.
Conclusions
18) The conclusion is aligned with findings but too generic. Please explicitly state:
- The final recommended structure (5 factors, 33 items).
- The importance of further cross-cultural and longitudinal validation.
- The value of reliable stress measures for guiding campus mental health policies.
Fewer recommendations
19) The manuscript contains many typographical errors and duplicated phrases (e.g., “virus' virus's severity,” “contamination factor cshould”) that must be corrected.
20) Terminology should be consistent: always use COVID-19 Stress Scale (CSS) after first introduction.
21) Abbreviations (CFA, EFA, RMSEA, CFI, etc.) should be defined at first mention.
22) References need careful editing: some are incomplete or incorrectly formatted (e.g., “Journal of Anxiety Diso ders”).
Author Response
|
|
|
|
|
|
|
|
|
|
|
|
|
|
|
|
|
|
|
|
Response to Reviewer 3 Comments
|
Thank you very much for taking the time to review this manuscript. Please find the detailed responses below and the corresponding revisions/corrections in track changes in the re-submitted files.
|
||
|
2. Questions for General Evaluation |
Reviewer 3 Evaluation |
|
|
Is the content succinctly described and contextualized with respect to previous and present theoretical background and empirical research (if applicable) on the topic? |
Can be improved |
|
|
Are the research design, questions, hypotheses and methods clearly stated? |
Can be improved |
|
|
Are the arguments and discussion of findings coherent, balanced and compelling? |
Can be improved |
|
|
For empirical research, are the results clearly presented? |
Can be improved |
|
|
Is the article adequately referenced? |
Can be improved |
|
|
Are the conclusions supported by the results presented in the article or referenced in secondary literature?
|
Yes |
|
|
Comments and Suggestions for Authors Thank you for the opportunity to review the manuscript entitled “Psychometric Properties of the COVID-19 Stress Scale in College Students”. The study aims to validate the COVID-19 Stress Scale (CSS) in a sample of U.S. college students.
The manuscript is well-structured, employs appropriate analyses (CFA/EFA), and provides useful insights into the scale’s psychometric performance. However, several methodological, theoretical, and editorial issues need to be addressed before the paper can be considered for publication.
Abstract The abstract is clear but too general. Please, report demographic details (mean age, gender distribution), briefly state the main modification (5-factor solution with item deletions), and replace vague statements (“authors provide future recommendations”) with more specific implications.
|
||
|
|
||
|
Response: Thank you. Demographic details have been added. The main modification has been stated. A statement of specific implications has been provided.
Introduction The introduction situates the study well within the pandemic context but contains redundancies and some awkward phrasing. Streamlining is recommended. Response: Thank you. This section has been revised for clarity and flow. The theoretical framing of stress should be strengthened by explicitly linking transactional stress theory (Lazarus & Folkman) and psychological measurement theory to the rationale for scale validation. Response: Thank you. This has been added as a reason for scale validation in the Introduction. Further discussion of stress appraisal and individual differences was also included/expanded later in the Discussion section.
The rationale for focusing on college students could be better justified using literature on student stress as a distinct phenomenon (e.g., academic stress, developmental stage). Response: Thank you. This has been expanded and more fully discussed in the Introduction section.
The framing could be strengthened by including more recent comparative international data on staff distress or burnout to highlight the gap this study fills. Indeed, burnout is a multi-component construct formed by different and complex size. It is also necessary to refer and comment on the authors' results in relation to the change that the theoretical framework on burnout has undergone in recent years. Furthermore, it is necessary to integrate the literature review, also referring not only to your own national context, but also international literature. Expanding the literature review beyond a national context by incorporating international perspectives would provide a more comprehensive understanding of the topic. Response: Thank you. A discussion of student burnout has been added to the Introduction, and is also explored in the Conclusions.
Methods The sample è adequately described, but response rate and recruitment beyond the introductory psychology course should be clarified. Were participants representative of the wider student body?
Response: Thank you. This has been clarified to include that the survey was available to all members of the student body, as well as faculty and staff.
The description of the CSS is adequate but incomplete. Please add: - Example items for each domain. - Reliability indices (Cronbach’s alpha) for subscales in this sample. - Whether the original English version was used or if any adaptation occurred. Response: Thank you. Sample Cronbach alphas are reported for both versions of the scale in text at the beginning of section 3.2 Data Analysis, and in section 3.4, Modified Five Factor Structure. A new table (Table 1) has also been added with descriptive item information for the CSS. The last sentence in section 2.2 Measures, now states that the English version of the measure was used with no adaptation.
Data Analysis CFA/EFA are appropriate, but the manuscript should clearly state: - Cut-off thresholds for RMSEA, CFI, TLI considered acceptable. - Whether measurement invariance across gender was tested (important given the gender imbalance in the sample). - How missing data were handled.
Response: Thank you. The cut-off thresholds were added to section 2.3, Analysis Plan, “To evaluate model fit, the following criteria for good fit were used: Chi-Square p > .05.CFI/TLI > .90, RMSEA < .05. A statement describing how missing data were handled was added to section 2.3, Analysis Plan, “listwise deletion used for missing cases)”. The cut-off thresholds were added to section 2.3, Analysis Plan, “To evaluate model fit, the following criteria for good fit were used: Chi-Square p > .05.CFI/TLI > .90, RMSEA < .05. The issue of measurement invariance was added to the Discussion “It should also be noted that this sample used in this study was very imbalanced in terms of gender. In addition to being an issue of sample generalizability, this imbalance made checking measurement invariance across genders not possible due to the small number of males included in the study.” Further discussion of this issue has also been added to that section.
Subsections “Participants,” “Measures,” “Data Analysis”) should be clearly separated for readability and to align with reporting standards.
Response: Thank you. These subsections have been added.
Results Results are presented in detail, but tables should conform to APA style (consistent decimal places, italicized statistical symbols).
Response: Thank you. This has been corrected.
The initial CFA model shows poor fit (CFI = .83, RMSEA = .09). The rationale for item removal (e.g., cash/mail items) needs to be more transparent — was this a purely statistical decision, or also conceptually justified (low relevance for young adults)?
Response: Thank you. This was clarified in section 3.2, Data Analysis,” however, three items can be considered a bare minimum for the definition of a psychometrically sound factor (Kline, 2000). Given the items only loaded well with each other (but to no other factors) and the somewhat antiquated nature of these issues in current culture, the decision was made to delete the items instead of retaining a sixth factor.”
Item-level results (means, SDs, factor loadings, inter-factor correlations) should be summarized more clearly to allow readers to evaluate construct validity.
Response: Thank you. Item level information appears in Table 1. Factor Loadings appear in tables 3 and 4.
Reporting effect sizes and reliability indices alongside model fit would strengthen the psychometric evidence.
Response: Thank you. Alpha if deleted reliability estimates have been added in table 1. Factor loadings (which appear in tables 3 and 4) are a measure of effect size in addition to the model fit indices. This seems to be the most efficient way to present the information.
Discussion The discussion acknowledges the modifications to the CSS but should better emphasize the novel contribution: this is one of the first large-scale validations in U.S. college students.
Response: Thank you. This has been added.
Comparisons with international CSS validations are too brief. A more systematic discussion is needed to explain why the contamination/cash/mail items failed in this context. Response: Thank you. The discussion of these issues related to items on the CSS instrument has been expanded in the Discussion section of the manuscript.
Practical implications should be expanded: - How can university counseling centers or student support services use the CSS? - What are the advantages of a shortened 33-item CSS for applied research and practice? Response: Thank you. This has been added to the Discussion.
Limitations should include: reliance on a convenience sample, absence of test–retest reliability, and lack of measurement invariance testing.
Response: Thank you. This has been added to the Discussion.
Conclusions The conclusion is aligned with findings but too generic. Please explicitly state: - The final recommended structure (5 factors, 33 items). - The importance of further cross-cultural and longitudinal validation. - The value of reliable stress measures for guiding campus mental health policies. Response: Thank you. These statements have been added to the Conclusions section and the value for guiding campus mental health policies has been expanded.
Fewer recommendations The manuscript contains many typographical errors and duplicated phrases (e.g., “virus' virus's severity,” “contamination factor cshould”) that must be corrected. Response: Thank you. This has been corrected.
Terminology should be consistent: always use COVID-19 Stress Scale (CSS) after first introduction. Response: Thank you. This has been corrected.
Abbreviations (CFA, EFA, RMSEA, CFI, etc.) should be defined at first mention. Response: Thank you. This has been corrected.
References need careful editing: some are incomplete or incorrectly formatted (e.g., “Journal of Anxiety Disorders”). Response: Thank you. This has been corrected.
|
||
|
|
||
|
|
|
|
|
|
||
|
|
||
|
|
||
|
|
||
|
|
||
Round 2
Reviewer 2 Report (New Reviewer)
Comments and Suggestions for Authors
Please check out the new table - Table 1. There are many mistakes in values of M and SD. SD cannot be bigger than SD. The rest of the paper is fine.
Author Response
Response: Thank you. The values have been thoroughly checked and accurately the M and SD for this dataset. The table presents means and standard deviations for items on a 0-4 point scale. The values are correct and indicate most items to have values around 1 or 2 with a variation of 1 point on average.
Reviewer 3 Report (New Reviewer)
Comments and Suggestions for Authors
I would like to thank the authors for their careful and thorough revisions. All my previous comments have been adequately addressed. I believe the changes have improved the quality and clarity of the manuscript. I have no further remarks.
Author Response
Response: Thank you for your time and constructive feedback on our manuscript.
This manuscript is a resubmission of an earlier submission. The following is a list of the peer review reports and author responses from that submission.
Round 1
Reviewer 1 Report
Comments and Suggestions for Authors
The study aimed to evaluate the psychometric properties of the COVID-19 stress scale, which was initially developed by Taylor et al. (2020). In the original scale, Taylor et al. (2020) collected their data during the COVID-19 pandemic lockdowns from March to April 2020, which may have been particularly critical for a questionnaire specific to a specific situation, such as a pandemic. It is valuable to evaluate new questionnaires in different sample groups; however, there are major concerns and questions related to the manuscript, which are pointed out below.
Major Points:
As mentioned earlier, the original questionnaire gathered data from people experiencing pandemic lockdowns, making it more appropriate to ask certain questions at that time. However, in this study, data were collected until 2022, so it is important to consider how the authors assess potential differences among individuals' worries after the pandemic and during the COVID-19 pandemic. Variations in the number of factors included in the questionnaire might relate to this issue. Therefore, the items excluded from the questionnaire focus on whether individuals are worried about catching the virus through handling money, transactions, or mail.
What are the full results of the statistics, such as chi-square, etc.? Only some of the fit indices were included in the manuscript, so it is not possible to evaluate the results of the analysis.
In the manuscript, the authors first introduce CFA followed by EFA. Generally, EFA precedes CFA. More information is needed on how the authors used EFA to investigate the model so that the reader can better understand the analysis.
One of my questions for the authors concerns the number of factors in their paper. In the original questionnaire article (Taylor et al., 2020), five factors are listed in the table of factor loadings, although the authors label six factors, with "danger" and "contamination" grouped under the same factor. As the authors mention on page 8, line 184, the loadings for items 19, 20, and 21 are very similar to those in the original article. However, the three items that the authors decided to exclude show very high loadings, so further analysis is needed to justify why these items are considered irrelevant, especially since in the original questionnaire, those excluded items were linked to a different factor related to worries about contamination. Why do the authors believe this factor is not relevant, and how do they explain these results through statistical analysis?
There are no other measures used in the research to assess validity. It would be another important factor. The authors' explanation for not conducting a validity assessment of the questionnaire, given that the aim was to evaluate the psychometric properties of a scale's reliability and validity, should be consistent.
It would be better to include information about the methods (i.e. number of participants) and results in the abstract, following the APA manual.
There is a typo on page 9, line 195, the excluded items 22,23 and 24, which was written as 25
Reviewer 2 Report
Comments and Suggestions for Authors
Abstract:
Line 14: Suggested edit to opening sentence in abstract: Many people experienced isolation and restricted behaviors due to the rapid onset of the COVID-19 pandemic in 2020.
Line 15: Re-word: However, investigations related to the psychological factors such as stress along with the danger of spread and contamination are scarce.
Line 18: Use “pandemic” as opposed to “virus”
Line 21: “Epidemic” may not be appropriate, “pandemic” or “acute health crises” may be better
Introduction:
Line 26: Need commas for “, among many other countries across the world,”
Line 28-30: Requires a citation
Line 33: Is there a more recent citation that Salleh, 2008?
Line 43-45: Is there a more recent citation that Cohen et. al, 1995?
Line 49: Same as above, is there a more recent citation than Delongis, et al., 1988?
Line 52: Same as above for Lazurus, 1966
Line 59: Same as above for Segerstrom and Miller, 2004
Overall the introduction provides a clear justification for this study. However, there have been studies that did investigate stress due to the COVID-19 pandemic, particularly over the first year. I would recommend looking at:
Eubank, J.M., Burt, K.G., & Orazem, J. (2021). Examining the psychometric properties of a refined perceived stress scale during the COVID-19 pandemic. [Special issue]. Journal of Prevention and Intervention in the Community. 49(2), 179-192. DOI: 10.1080/10852352.2021.1908873.
Along with their follow-up works:
Eubank, J.M., Oberlin, D.J. II, Orazem, J., & Pegues, M.M. (2024). The impact of COVID-19 on college student leisure time physical activity, sedentary behavior, and stress in New York City. Discover Psychology 4(88). DOI: 10.1007/s44202-024-00200-y
Burt, K.G., Eubank, J.M., & Orazem, J. (2022). Female black, indigenous, and students of color demonstrate greater resilience than other students during a global pandemic. Race Ethnicity and Education. DOI: 10.1177/27526461221105094.
Burt, K.G. & Eubank, J.M. (2021). Optimism, resilience, and other health-protective factors among students at a New York City Hispanic-serving institution during the COVID-19 pandemic. Journal of Effective Teaching in Higher Education. 4(1), 1-17. DOI: 10.36021/jethe.v4i1.206.
Materials and Methods:
Lines 127-131: At some point, authors should include an IRB approval statement.
Lines 133-140: This manuscript would benefit from the authors giving more background about how this scale was developed, where the questions came from, and if it has been validated in any other studies that have been published. If so, include the population and results.
Results:
Lines 143-144: The survey was open for approximately 1 year and 4 months. Isolation, distancing, safety, and many other practices changed throughout that timeframe
Discussion:
Line 218-220: I would suggest editing this sentence as it can lead to confusion on which study the authors are talking about, the current one or Taylor and colleagues.
Line 218: In lines 143-144: The authors state that the survey was available from November 2020 to March 2022 (1 year and 4 months). The authors then state that the study was conducted during the early stages of the pandemic. I would recommend re-stating that as almost 1.5 years later may not be viewed as the “early stages.”
Overall, the discussion addresses the issues within the psychometric properties of the CSS. However, I would recommend comparing this to other studies conducted on stress related to the pandemic, which could add more breadth to the discussion section.
I would suggest reviewing the manuscript for proper sentence structure along with more recent citations (no older than 10 years).